# The coevolution of the firm and the product attribute space

**César García-Díaz** [1] *, **Gábor Péli** [2,3], **Arjen van Witteloostuijn** [4,5]

**1** Department of Business Administration, Pontificia Universidad Javeriana, Bogotá, Colombia, **2** Centre for Social Sciences, Hungarian Academy of Sciences Centre of Excellence, Budapest, Hungary, **3** Institute of Social and Communication Sciences, Károli Gáspár University of the Reformed Church in Hungary, Budapest, Hungary, **4** School of Business and Economics, Vrije Universiteit Amsterdam, Amsterdam, The Netherlands, **5** Faculty of Business and Economics/Antwerp Management School, Antwerp, Belgium

\* ce.garciad@javeriana.edu.co

## Abstract

Traditionally, firm competition has been studied in contexts where the dimensionality of the product attribute space is given, and firms deploy their strategies constrained by this space. However, firms may exert influence on the local structure of the product attribute space by offering product variants with new attributes. As a result, the geometry of the product attribute space would change endogenously through firms' actions, and this emergent new geometry modifies the conditions for subsequent firm behavior. By focusing on this interplay between actors and conditions, we explore the co-evolution of the firm and the product attribute space. Through a multi-variant Cournot competition framework, we develop a computational model in which firms invest to differentiate their products from other variants, but as minimally as possible so that demand from closely similar existing variants can be stolen. We introduce the fraction dimensionality of the attribute space as our critical independent variable, to reflect saturation of the space with product varieties. The simulation reveals that while new product variants are typically introduced by firms with scale economies, their performance gap with firms without scale economies reduces as fraction dimensionality increases. This indicates that space geometry evolution may favor small-scale players, even when their large-scale competitors are the driving force behind attribute space changes.

## 1 Introduction: Commodity space evolution

Attribute spaces display how demand for, and supply of, organizational services distributes over a number of attributes, or dimensions, that characterize the offerings [1]. As the number of product attributes taken into account by economic actors is a natural number, we straightforwardly associate the complexity of the attribute space with its positive integer dimensionality. The attribute space can be a commodity space spun by product characteristics that prospective customers evaluate [1, 2, 3], a market place that combines product design and product quality features [4], or a political issue space within which political representation is offered [5, 6]. In organization science, this can be a Blau-space [7] dimensioned by people's

**Data Availability Statement:** Our work is a computer simulation study, which does not contain real data. The computer code (NetLogo) and corresponding documentation can be found in the following public repository: (https://www.comses.

net/codebases/30b41925-c56d-4428-b812-
4351563cddcf/releases/1.0.0/).

**Funding:** This project has been financially
supported by the "Vicerrectoría de Investigación /
Facultad de Ciencias Económicas y
Administrativas" of Pontificia Universidad
Javeriana (Project ID: 9546; Dr. César García-Díaz).

**Competing interests:** The authors have declared
that no competing interests exist.

socio-demographic characteristics that motivate their choices with respect to organizational offerings [8, 9, 10, 11].

Empirical research in the management and marketing literatures argues that increasing the number of product variants may increase the demand for the firm's offerings [12], hence boosting this firm's performance [13], being a key instrument in the competitive battle. Extending product lines proves to be an important strategic competitive move, except if the extension is excessive [14, 15, 16]. So, as this holds for all firms in a given marketplace, to increase their viability, these firms get involved in an innovation race. Doing so, they may either offer new products residing in positions of the extant attribute space or embark on developing new product attributes, so inducing changes in the space (Fig 1).

Most extant research assumes that the dimensions of attribute space are exogenously given and consider that as the first modeling option. A less frequently visited modeling option is to take that attribute space dimensionality changes endogenously as a consequence of firms' actions. In spite of the relative scarcity of work taking that second modeling route, economic and organizational research have already disclosed a number of mechanisms regarding how changing integer dimensionality impacts market outcomes. For instance, the competition framework of [18] demonstrates why firms might seek for maximum product differentiation along a single dimension whilst following a minimum differentiation strategy along all others. [19] pointed out that increasing dimensionality can open demand pockets, which are safe havens for small-niche firms. [17] explored the strategic implications of developing new

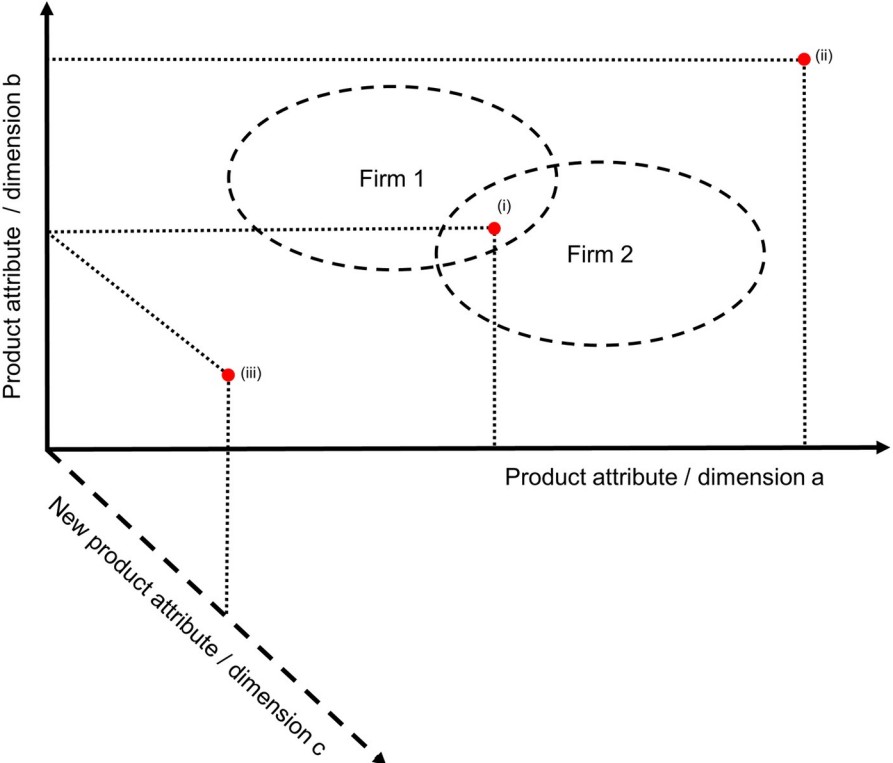

**Fig 1. Example of strategic alternatives for new competitor's positioning.** A space with two existing product attributes (a and b) and two established firms (firms 1 and 2) is depicted. The figure exemplifies three different strategic positioning alternatives for a new competitor: (i), (ii), and (iii). A new market rival could opt for: (i) competing at locations where firms are already established (dashed lines indicate Firm 1 and Firm 2's niches); (ii) occupying existing but empty spaces; or (iii) developing new product attributes (e.g., attribute c). This figure is our elaboration, but inspired by that of [17].

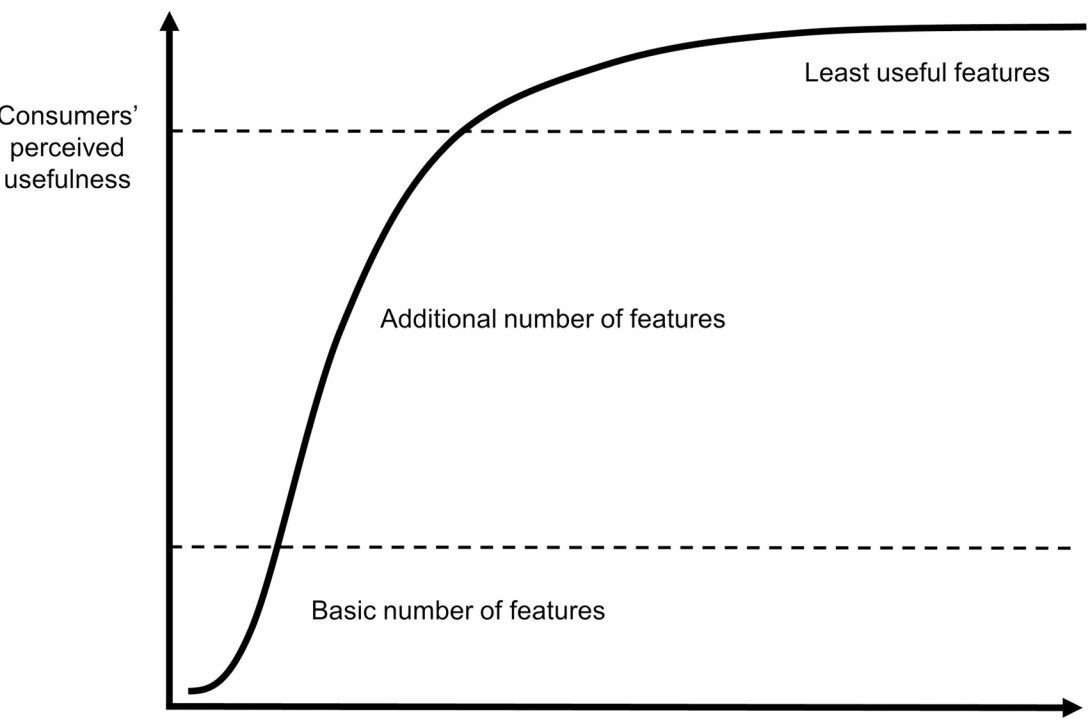

**Fig 2. Increasing perceived usefulness for consumers by adding features (attributes) to products.** Attributes add value to consumers at an increasing rate up to a certain point. Further additions add value at a decreasing rate. Beyond a certain number of features, additional ones might not add any value. This figure is our elaboration, but inspired by that of [22].

product attributes relative to other market positioning strategies. [20] computed how a monopoly firm's optimal market area diameter changes with changing space dimensionality. [21] pointed out that small firm subsistence can also be a result of the substantial enlargement of market peripheries, the latter being a by-product of increasing space dimensionality.

However, the proliferation of product attributes–or spatial dimensions–is likely to be limited for two reasons, at least. One is that human cognition can hardly consider more than a handful of alternatives. In line with this observation, the rise of new product variants enhances their perceived usefulness at a decreasing rate (Fig 2). Another limitation is that the introduction of a new product attribute, including its acceptance by prospective customers, takes time and requires investment. Therefore, the prevalent way of horizontal product differentiation is keeping the number of spatial attributes intact and offering new products variants by positioning away from other products along the extant axes.

In this paper, we offer a modeling alternative exempt from this limitation on the number of product dimensions. We propose an approach in which a new form of dimensionality change reflects market structuration processes even when the integer dimensionality of the space is constant. We apply the concept of *fraction dimensionality*, a measure that reflects the degree of market saturation with product variants. Higher market saturation with product varieties means having offerings at more locations of the attribute space. This saturation concept (cf. Section 2) does not reflect the amount of total demand satisfied directly, although normally more demand is satisfied with adding new product variants [12]. Fraction dimensionality is also a way to capture the degree of product heterogeneity in a market (or product space

complexity; see [13]). The introduction of a new product variant increases the fraction dimensionality of the attribute space by activating an empty location. This process establishes newly occupied product patches or extends existing conglomerates of similar products. The focal issue in our research is exploring the differential impact of increasing product variety saturation on firm strategy success.

Some of the appearing product variants may be known by customers, because they have already been offered earlier, and so the target audience has memories of their properties. An example is the re-introduction of alcoholic beverages to legal markets after the American prohibition period. In this paper, however, we explore contexts when such product knowledge either never existed or already has been washed out from collective memory. In our approach, each new product variant first has to be introduced to the target audience, and next its existence and usefulness have to earn a taken-for-granted status [23, 24]. The act of introducing a yet unknown variant to a market also impacts upon the tissue of the attribute space: now a meaningful product resides where just a 'hole' had been before. Thus, the fraction dimensionality measure can be applied to explore endogenous economic processes taking place in competitive markets, as well as to examine their societal embedding. On the one hand, fraction dimensionality increases when a new offering appears on the market. On the other hand, this dimensionality change materializes when market actors become familiar with the new offering.

The rest of the paper explores the consequences of that sort of product introduction for the relative success of two firm types: those with and without scale economies. The next Section 2 introduces the concept and use of fraction dimensionality. Section 3 presents our agent-based simulation model with competing firms featuring different cost structures, gradually updating their quantity choice strategies and saturating the attribute space with their offerings. Section 4 displays the results. Section 5 concludes.

## 2 Fraction dimensionality

The attribute space is spun by $k$ axes ($k$ is an integer number) so that products are $k$-vectors, each component standing for a value variant along the corresponding product attribute. Each axis has a finite number of attribute values, so possible product variants are modeled as discrete locations (i.e., cells) in a finite $k$-space segment. No producer or buyer can perceive differences between products within the same cell. This integer $k$-dimensional space segment serves as a frame at the simulations to come. Market saturation with new product variants will be characterized by the changing *fraction dimensionality* of this frame space.

We use Mandelbrot's *similarity dimension* concept [25] to define fraction dimensionality. Fraction dimensionality has already been applied to assess habitat loss and habitat fragmentation in the ecological literature (cf. [26]). It has also been used in the exploration of properties of quantity time series in dynamic Cournot competition with differentiated goods [27], in price time series of duopoly Bertrand competition [28], and in understanding the performance of large and small-scale firms over patchy demand landscapes in market evolution [29]. Assume a $k$ integer dimensional Euclidean space segment with $m$ scale elements per axis (frame space). Without restricting generality, we can assume unity distance between neighboring variants, thus rendering the frame space a $k$-dimensional hypercube composed of $m^k$ cells (Considerations of the case of unequal number of scale elements per axis are presented in the supplementary material, S1 File). A cell is called active when the corresponding product variant is offered at the given moment. Then, the fraction dimensionality *DIM* of the space with $V$ number of active cells from the $m^k$ total is defined as:

$$DIM = \frac{ln\ V}{ln\ m}. \tag{1}$$

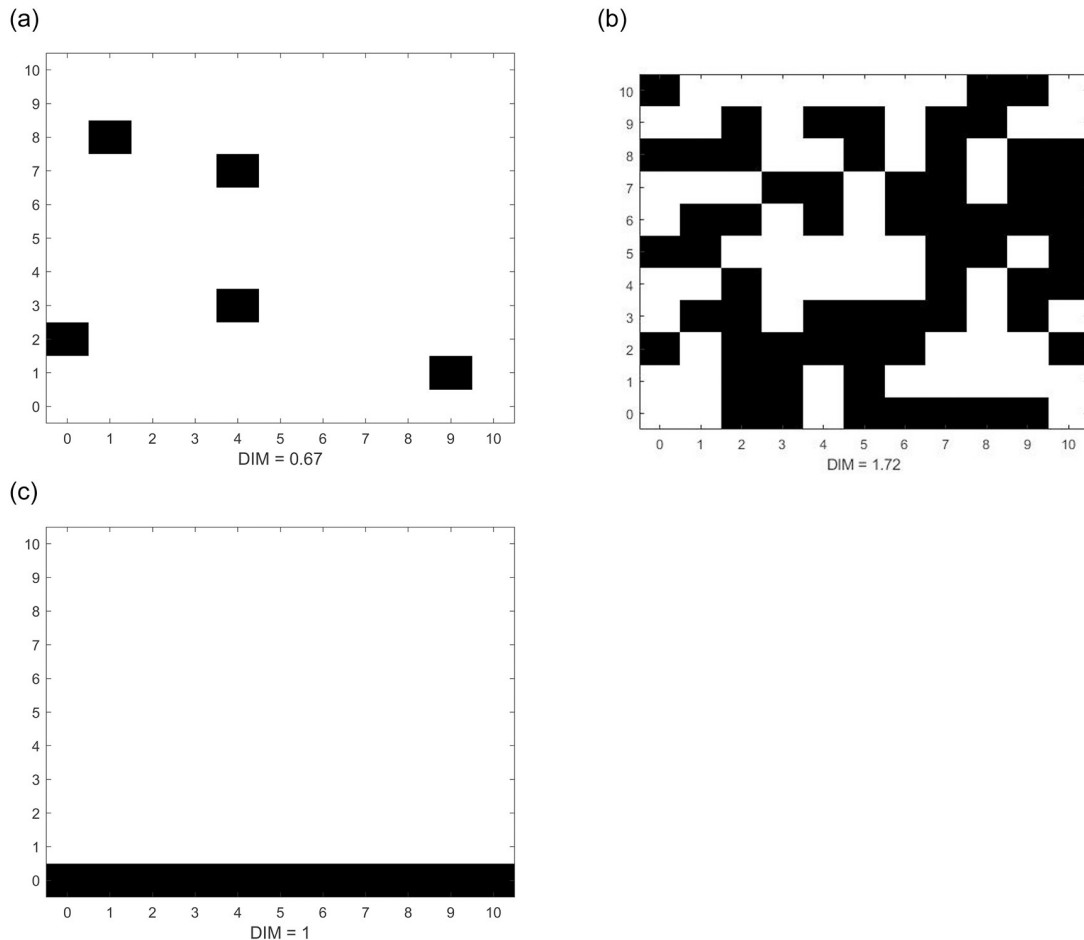

**Fig 3. Three possible states of a two-dimensional frame space.** Each attribute has 11 possible variations (from 0 to 10). Therefore, there are $11^2 = 121$ cells. Black squares represent active cells. Fraction dimensionality is (a) 0.67, (b) 1.72 and (c) 1.00 respectively.

Fig 3 illustrates three different degrees of space saturation and their corresponding fraction dimensionality values.

Note that (1) yields integer dimension $k$ as a special case when all $m^k$ cells are active (full saturation):

$$DIM = \frac{ln(m^k)}{ln\ m} = k. \tag{2}$$

Accordingly, the fraction dimension in Fig 3 would reach its maximum value 2 in case of having a 'black box' with 121 active cells. Another advantage of definition (1) is that it may yield the corresponding integer dimension value for fully saturated subspaces as well–for example, $DIM = 1$ for the horizontal line in Fig 3c. Similarly, $DIM = 2$ fraction dimensionality would apply for all vertical and horizontal planes with all cells occupied in $k = 3$ dimensional frame spaces. These properties of the Mandelbrot formula explain why we decided not to choose the simpler percentage measure to represent market saturation. For example, the percentage measure value of the linear subspace in Fig 3c would be 0.09, not 1 as expected.

Definition (1) is not sensitive to distribution of product offerings in space; its value only depends on the active cell count. Consequently, this measure can be applied to contexts without respect to scale type (nominal, ordinal, interval or ratio scale). Still, new products do not appear in the space randomly. Offerings tend to coalesce into spatial patches. Our simulation model reflects this observation by introducing new product variants in the vicinity of existing ones (cf. 3.4).

## 3 Firm competition in evolving attribute spaces

Next, we describe an agent-based computational model of firm competition, in which firms continuously update their quantity offerings, with the space endogenously changing as a consequence of firms' actions. We use a traditional quantity-based competition approach (Cournot competition). Basic Cournot models are one-shot and game-theoretic in nature, inspecting market equilibria as a result of firms' responses, according to quantity-based strategies. However, models *à la* Cournot have also been used in evolutionary settings to study, for example, reinforcement learning effects [30], implications of social and individual learning [31], impact of competing behavioral rules [32], effects of firm entry and exit on industry dynamics [33, 34], and the influence of product differentiation on equilibrium stability [35]. Much recent work takes Cournot competition in dynamic settings as the steppingstone to study nonlinear behavior and equilibrium stability (e.g., [36, 37, 38, 39]).

Below, we describe the rules of the competition unfolding in the patchy landscape of offerings. The baseline example is two-firm competition, as introduced in Subsection 3.1. In Subsection 3.2, this example is extended to a model of *multi-variant* Cournot competition over a patchy landscape of offered product variants. In Subsections 3.3 to 3.5, we display the rules of firm behavior within this framework.

### 3.1 Baseline: Two-firm Cournot competition

As a building block of the full model, let us consider a numerical example with two firms. Firms 1 and 2 compete in a market offering a single product variant (see scenario A in Fig 4). Price $P$ is a linearly decreasing function of demand, $P = a − bQ$, where $a$ and $b$ are the intercept and slope of the price function, respectively, and $a/b$ is aggregated demand. $Q$ is the total quantity sold by the two firms. Firms have an identical $C(q) = cq$ cost function, where $q$ is the quantity offered by each firm and $c$ is the unit cost of production. Let us define $q_i$ as the quantity

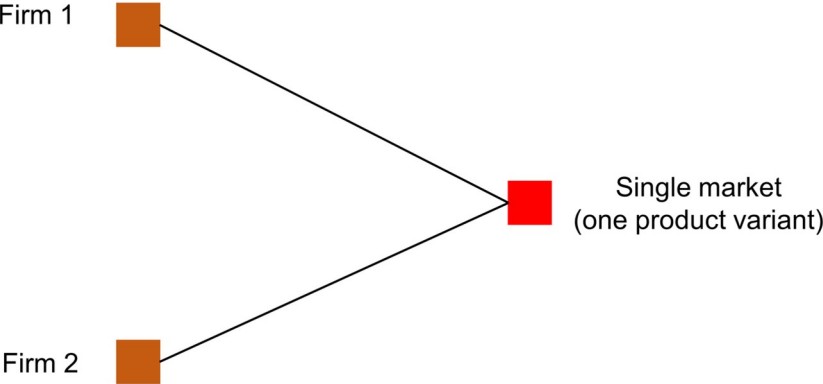

**Fig 4. Scenario A: Quantity competition with a single product variant.** Markets are represented by their offered product variants. Edges connect firms with markets denoting in which product variants a given firm operates. In scenario A, there is a single market with two firms.

produced by firm $i$. Accordingly, the $\pi_i$ denote the profit function of firm $i$, which is

$$\pi_i = (a - bQ)q_i - C(q_i), i = 1, 2. \tag{3}$$

Let us also define $\pi_{A,i}$ as the profit that optimizing firms get under scenario A, and $q_i^*$ as firm $i$'s optimal quantity. If $a = 100$, $b = 1$, and $c = 1$, then each firm maximizes profits when $q_i^* = 33$, $i = 1, 2$. This makes $\pi_{A,1} = \pi_{A,2} = 1{,}089$ under scenario A with a single product in place. Let us now assume that Firm 2 is able to launch a new product variant.

Assume that the cost of opening a new variant is $K$, and the cost of handling the complexity of more than one variant is $N$. The $N$ cost accounts for the scope diseconomies firms incur when serving demand from several markets (niche spanning costs; see García-Díaz et al., 2008). For the sake of demonstration in the example, we set $K = N = 1$.

Consumers may be lured to consume the new variant. For that, we assume that consumers are willing to switch to a new variant with a mobility rate $\vartheta = 0.2$. That is, 20% would be willing to consume the new variant. Notice that if $a/b$ is aggregated demand, and if the intercept value $a$ is kept constant, the new variant has a potential demand of $\vartheta\left(\frac{a}{b}\right) = a/\left(\frac{b}{\vartheta}\right)$, so the slope of the price equation for the second variant would be $b/\vartheta$ (that is, the market $\vartheta\left(\frac{a}{b}\right)$ clears when the price equals zero). A similar change happens to the existing variant, with its demand becoming $(1 - \vartheta)\left(\frac{a}{b}\right)$. Of course, the price equation for the second variant may change in different ways (both the intercept $a$ and the slope $b$). We assume that the intercept is fixed because: (i) to obtain positive quantities, $a$ has to be set much larger than unit cost $c$, so a decrease of $a$ for subsequent product variants is not desirable for modeling purposes; and (ii) an increase of $a$ for subsequent product variants would likely generate higher prices, rendering the investigation of whether firms would open new variants trivial as new variants usually charge higher prices. Thus, a more restrictive way of exploring the model is, first, assuming $a$ fixed while $b$ is recalculated according to the demand allocated to the product variants and, second, assuming that the market is cleared when the price equals zero (we also adopt these assumptions in the full computational model).

Knowing the above, Firm 2 would open a new variant if its monopolistic profit in the new market will offset the sum of investment $K$, the additional cost $N$, and the profit loss that results from redistributing the market between two product variants instead of one (see scenario B in Fig 5).

Assuming that Firm 1 will not change its production quantity, Firm 2 will introduce a new product variant if the profit under this second scenario B with a new variant is higher than in case of a single variant (scenario A, $\pi_{A,2} = 1089$). Let $q_{2,1}$ and $q_{2,2}$ denote the quantities that Firm 2, respectively, will allocate to variants 1 and 2 under scenario B. Instantiating these quantities to Eq (3) gives the profit under scenario B for Firm 2, $\pi_{B,2}$:

$$\pi_{B,2} = \left[ \left( a - \left( \frac{b}{1 - \vartheta} \right)(q_{2,1} + q_1^*) \right) q_{2,1} - cq_{2,1} \right] + \left[ \left( a - \left( \frac{b}{\vartheta} \right)q_{2,2} \right) q_{2,2} - cq_{2,2} \right] - (N + K). \tag{4}$$

The quantities that maximize Firm 2's profits are $q_{2,1} = 23.1$ and $q_{2,2} = 9.9$, making the profit under scenario B higher: $\pi_{B,2} = 1155.06 > 1089 = \pi_{A,2}$. So Firm 2 will opt for introducing a new product variant.

## 3.2 Multi-variant Cournot competition

We extend the above model to an arbitrary number of firms. Firms decide what production quantities they allocate to the variants in their product portfolios. Competition is akin to *networked Cournot competition* [40, 41]: firms not only decide how many product variants to serve, but also how much to allocate in each of them. The above-mentioned studies illustrate

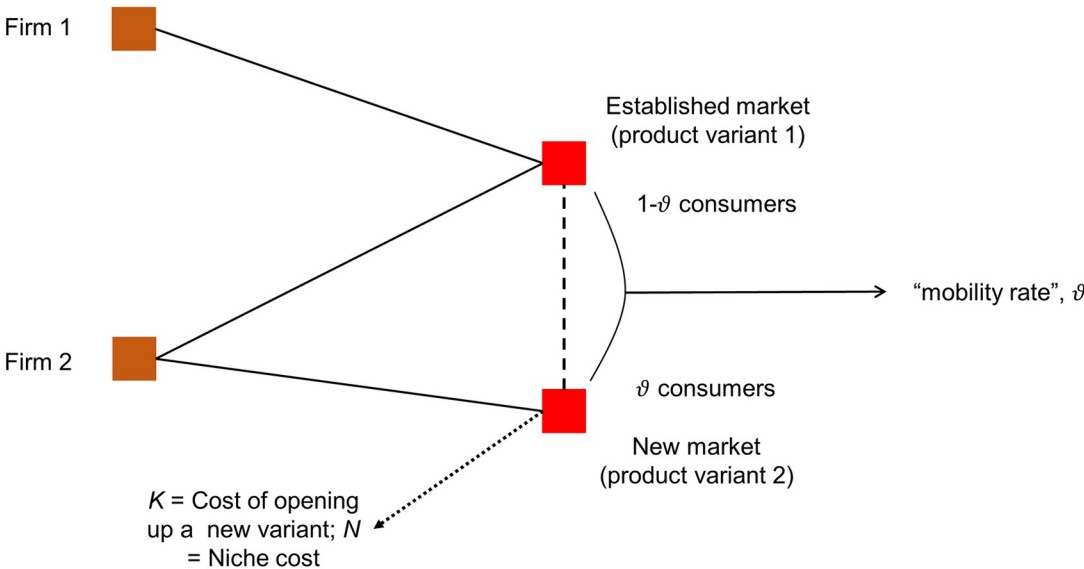

**Fig 5. Scenario B: Quantity competition when a firm faces the possibility of opening a new product variant.** Markets are represented by their offered product variants. In scenario B, two Firms 1 and 2 operate in the established market, while only Firm 2 is present in the new one. New consumers are dragged into the new market with mobility rate $\vartheta$.

game-theoretical approaches to competition in bipartite networks of firms and markets. There are important antecedents to the networked Cournot competition setting in the literature. For instance, [42, 43] and [44] study how R&D networks affect firms' marginal cost of production, which in turn affect optimal quantity outcomes in the market. Nonetheless, unlike all this prior work, our model differs in the way network structures affect Cournot competition outcomes. Our work portrays an evolving bipartite network of firms and product variants. The market has an evolving number of product variants with a constant total demand $M$ distributed equally over cells.

New product variants emerge as a consequence of firm actions. Firms develop new product variants according to two criteria: (i) they aim at expanding their offerings to some new cells, expecting to be, at least temporarily, the only supplier of the pertaining product variant; and (ii) the new product variant has to be located close to existing variants (e.g., [45]) so that suppliers of the former can possibly steal consumers from the latter on the basis of product similarity and substitution. Thus, the likelihood of the act of offering a new product variant depends on the firm's occupation of existing variants: firms try to tamper with the competition they face at crowded cells by activating new cells. Total market demand is given, so the demand for a new product variant depends on the willingness of consumers of existing variants to switch.

As said, we apply the concept of bipartite network. The nodes of bipartite networks are divided into two sets, and only nodes from different sets are connected. The model represents firms and their product variant offerings as displayed in Figs 6 and 7. Fig 6 illustrates an example of an evolving bipartite network. Fig 7a shows an example of a possible mapping between firms and product variants. Solid lines indicate links between firms and variants. Axes in Fig 7b represent values of product attributes. In this two-dimensional example, a given combination $(x, y)$ of product attributes corresponds to a given product variant in attribute space. Firms open new variants that are minimally differentiated (they coincide with one of the

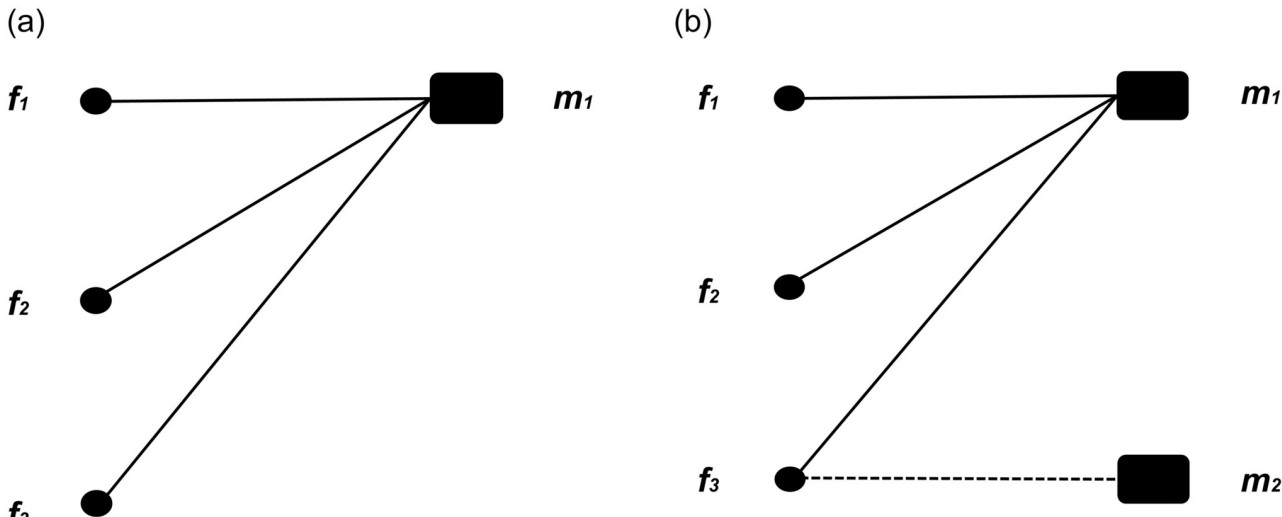

**Fig 6. The market as an evolving bipartite network, mapping firms to product variants: $f_1$, $f_2$ and $f_3$ represent firms, while $m_1$ and $m_2$ are product variants.** In panel (a), all firms compete with the same product variant. In panel (b), firm $f_3$ opens a new product variant $m_2$ (dashed line), so that part of the consumers of $m_1$ are dragged into $m_2$.

attributes, but differentiate with the other). Dashed lines in Fig 7b indicate how variants are interrelated in attribute space.

## 3.3 Costs and profits

Our simulations aim at exploring how the introduction of product varieties impacts firms with different characteristics. We performed sensitivity analyses along three focal variables: niche-

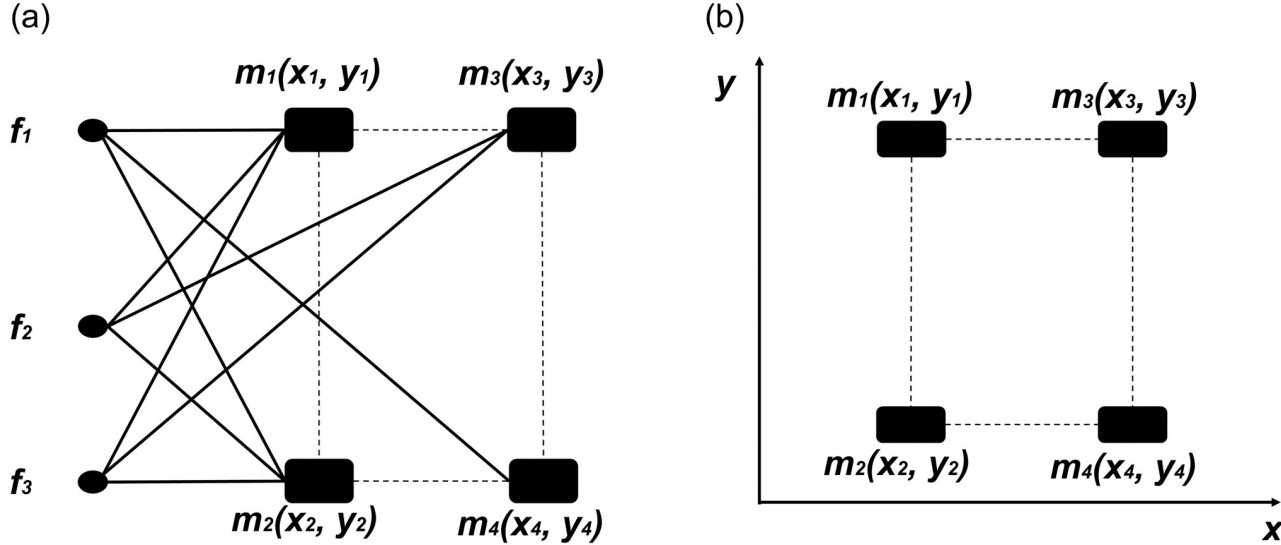

**Fig 7. An example of a competition network mapped onto a set of product variants.** Solid lines [panel (a)] are mappings from firms to products. Each firm serves a subset of the available products $\{m_1, m_2, m_3, m_4\}$. Product nodes are embedded into a $k$-space [46], with $x_i$ and $y_i$ denoting product coordinate values. Axes [panel (b)] represent values of product attributes. Dashed lines [panel (b)] indicate that two products are only differentiated in one of the attributes. Products $m_1$ and $m_3$ are horizontally differentiated along their $x$ attribute; the same holds for $m_2$ and $m_4$.

spanning costs (i.e., the costs of maintaining a portfolio of product variants), the introduction costs of new product variants, and the total number of firms competing for a given market demand. This variable choice necessitated having other model parameters constant, for the time being, in order to keep model complexity at bay.

During each simulation run, we had a constant overall demand $M$ and a constant population of an $F$ number of firms that fall into two characteristically different types: those with scale economies ($\alpha$-type) and those without scale economies ($\beta$-type). For the sake of analytical convenience, we abandon time index $t$ from the formulae, for the time being. $n_{\alpha,j}$ and $n_{\beta,j}$ denote, respectively, the number of firms per type that offer product variant $j$. $\beta$-type firms have a constant unit production cost $c_{\beta,j}$ at variant $j$; $\alpha$-type firms are able to reduce unit production costs as production volume increases by exploiting scale economies. Their $c_{\alpha,j}$ realized unit production cost at variant $j$ depends on the $q_{i,j}$ production quantity:

$$c_{\alpha,j} = c_{\beta,j} - e_j q_{i,j}. \tag{5}$$

Maximal cost $c_{\beta,j}$ in (5) is drawn from a uniform probability distribution. The $e_j > 0$ value is $j$'s unit cost reduction impact per production unit. To avoid negative $c_{\alpha,j}$ values in (5), we establish a minimum unit cost $c_o$, so that $c_{\beta,j} - e_j M \geq c_o$, with $M$ standing for total demand. This implies that that $c_{\beta,j} \geq c_o + e_j M$. We opt for setting $c_o = 0$.

Firm $i$ calculates its profit from variant $j$ as follows:

$$\pi_{i,j} = P_j q_i - c_{r,j} q_{i,j}. \tag{6}$$

Coefficient $c_{r,j}$ represents the unit production cost; $r$ denotes firm type ($\alpha$ or $\beta$). Assuming $n_j$ firms offering product variant $j$, the price of $j$ is:

$$P_j = a - b_j Q_j = a - b_j \sum_{i=1}^{n_j} q_{i,j}, \tag{7}$$

where $Q_j$ is total quantity at variant $j$, and the non-negative $a$ and $b_j$ are the price equation intercept and slope values, respectively. Firms may get involved in direct competition for the demand associated with each variant. Since the price is set by the market, firms choose quantities to maximize their profits (à la *Cournot*).

Denoting the total number of firms in variant $j$ as $n_j = n_{\alpha,j} + n_{\beta,j}$, and knowing that $Q_j = \sum_{i=1}^{n_{\alpha,j}} q_{i,j} + \sum_{i=n_{\alpha,j}+1}^{n_{\alpha,j}+n_{\beta,j}} q_{i,j}$, the total quantity firms offer from variant $j$ is:

$$Q_j = \frac{n_{\alpha,j} - \left[\frac{(2e_j - b_j)}{b_j}\right] n_{\beta,j}}{b_j n_{\alpha,j} - (2e_j - b_j)(n_{\beta,j} + 1)} \left(a - c_{\beta,j}\right). \tag{8}$$

The implying quantities per firm type, $Q_{\alpha,j}$ and $Q_{\beta,j}$, are:

$$Q_{\alpha,j} = \frac{n_{\alpha,j}}{b_j(n_{\alpha,j} + n_{\beta,j} + 1) - 2e_j(n_{\beta,j} + 1)} \left(a - c_{\beta,j}\right), \tag{9}$$

and

$$Q_{\beta,j} = \frac{\left[\frac{(b_j - 2e_j)}{b_j}\right] n_{\beta,j}}{b_j(n_{\alpha,j} + n_{\beta,j} + 1) - 2e_j(n_{\beta,j} + 1)} \left(a - c_{\beta,j}\right). \tag{10}$$

Quantities are the same for each firm within a type. So if firm $i$ is of $\alpha$-type, its production level would be $q_{i,j} = Q_{\alpha,j}/n_{\alpha,j}$; if it is of $\beta$-type, then $q_{i,j} = Q_{\beta,j}/n_{\beta,j}$. Notice that if $n_{\alpha,j,t} = n_{\beta,j,t} = 1$, $b_j = 1$ and $e_j = 0$, then the production quantities are $(a - c_{\beta,j})/3$, which is the optimal production

volume in a two-firm, one-shot Cournot model with homogeneous production cost, $c_{\beta,j}$. Restricting quantities to positive values, Eqs (9) and (10) imply that $e_j < b_j/2$ and $a - c_{\beta,j} > 0$ should hold. The detailed calculations are in the supplementary material, S1 File [Equations SP1 to SP13].

Firms can have offerings in several product variants simultaneously. They may also open new ones if the one-time costs of introducing a new variant is lower than the expected benefit earned by doing so. Firms also face scope diseconomies in the form of *niche width costs*: the larger the distance between the offered product variants of the same firm, the higher these niche spanning costs become. Serving very heterogeneous audiences is expensive [29, 47], and the gains from scale economies (vertical expansion) may be offset by the losses coming from operating in broad niches with diverse consumer preferences (horizontal expansion; see [48]). $N$ is the niche width distance unit cost.

Now we add time index $t$ to the model. The niche of firm $i$ is defined as its set of product variants $H_{i,t}$ at time $t$. Let $s_{H_{i,t}}$ denote the niche width of firm $i$, defined as the largest *block distance* between any two variants' in its $H_{i,t}$ niche. The *block distance* of two locations is the aggregate of their distances along attributes. Applying other distance measures (e.g., Euclidean distance or the largest of the shortest paths among the network of variants) is also possible. But since attributes have different units of measure, having the firm computing the pairwise product variant dissimilarity *per attribute*, and figuring out the aggregated largest distance of its served variants, is a more reasonable rule (a similar approach has been used to compute preference dissimilarity in voting systems; see [49]).

The $Ns_{H_{i,t}}$ total niche width cost depends on the value of this supremum distance. Then, firm $i$'s total profit is the sum of the profits earned at each product $j$ minus its total niche width cost:

$$\pi_{i,t} = \sum_{j \in H_{i,t}} \pi_{i,j,t} - Ns_{H_{i,t}}. \tag{11}$$

Firm $i$'s total production quantity is the sum of the quantities offered at variants $j$:

$$q_{i,t} = \sum_{j \in H_{i,t}} q_{i,j,t}. \tag{12}$$

The intensity of competition at a given product variant can be affected by another firm that decides to invade the cell of this variant or to introduce a new variant in its vicinity. The creation of new variants triggers the redistribution of total demand across existing ones. In sum, competition is *networked* or *nested* across interrelated product variants.

### 3.4 Expansion dynamics

At time $t$, firm $i$ decides whether to

 i. keep its currently served variants,

 ii. open up a new variant (location is selected randomly from neighboring positions),

 iii. expand into an existing neighboring variant, or

 iv. abandon one of its served product variants.

Alternatives (i-iv) are considered simultaneously at each simulation step, and the firm chooses the one with the highest expected profit.

If firm $i$ keeps its current product niche [choice (i)], its $\tilde{\pi}'$ profit expectation for the next step is:

$$\tilde{\pi}\prime_{i,t+1} = \sum\nolimits_{j \in H_{i,t}} \tilde{\pi}_{i,j,t+1} - Ns_{H_{i,t}}. \tag{13}$$

Superscripts, $''$, $'''$, and $''''$ in the forthcoming profit formulas denote respective actions i, ii, iii, and iiii the firm takes. Term $\tilde{\pi}_{i,j,t+1}$ in Eq (13) is expected profit at time $t + 1$ from variant $j$. There are several alternatives to compute profit expectations, including sophisticated demand redistribution procedures [50]. Yet, a simple heuristic can be a reasonable firm choice when facing complex and uncertain environments (cf. [51, 52]). We assume that firms expect to obtain at least the same profit at variant $j$ that they had obtained in the previous time step, so avoiding the complex and highly uncertain computation of expected demand redistributions influenced by rival firms. This heuristic constitutes a conservative way of proceeding: firms optimistically take the recent profit as benchmark at deciding to expand their product niche to an existing product variant, to create a new variant or to drop an old one.

Total market demand is fixed. When a new product variant appears, its demand surplus decreases demand at all active product cells equally. Thus, after the creation of a new product variant [choice (ii)], the firm expects to gain:

$$\tilde{\pi}\prime\prime_{i,t+1} = \sum\nolimits_{j \in H_{i,t}} \tilde{\pi}_{i,j,t+1} - Ns_{H_{i,t} \cup \{d\}} + \tilde{\pi}_p - K. \tag{14}$$

Again, $\tilde{\pi}_{i,j,t+1}$ is expected profit at time $t + 1$ from variant $j$; $s_{H_{i,t} \cup \{d\}}$ is the niche width of firm $i$ after including new product variant $d$. $K$ is the one-time cost of opening a new cell; and $\tilde{\pi}_p$ is the expected monopolistic profit at the newly targeted variant. The monopoly quantity values depend on firm type. Since the location of the new variant is irrelevant here, we abandon product variant indices. Being $\tilde{c}_\beta$ the expected unit cost of the targeted variant, and $a$ and $b$ the price function parameters, $\beta$-type firms' monopolistic profit is calculated as:

$$\tilde{\pi}_p = (a - bq^*)q^* - \tilde{c}_\beta q^*, \tag{15}$$

where production quantity is $q^* = (a - \tilde{c}_\beta)/2b$.

Denoting $\tilde{e}$ as the expected scale factor of the targeted variant, an $\alpha$-type firm's monopolistic profit is:

$$\tilde{\pi}_p = (a - bq^*)q^* - (\tilde{c}_\beta - \tilde{e}q^*)q^*, \tag{16}$$

and $q^* = (a - \tilde{c}_\beta)/(2(b - \tilde{e}))$. Both $\tilde{c}_\beta$ and $\tilde{e}$ are randomly generated. Choice (iii) is to expand into an active cell: i.e., to one with an incumbent product variant $j$. Then, firm $i$'s profit is:

$$\tilde{\pi}\prime\prime\prime_{i,t+1} = \sum\nolimits_{j \in H_{i,t}} \tilde{\pi}_{i,j,t+1} - Ns_{H_{i,t} \cup \{d\}} + \tilde{\pi}_{i,d,t+1}. \tag{17}$$

Term $\tilde{\pi}_{i,d,t+1}$ is the profit the firm expects when entering variant $d$. To decide whether to do so, the firm checks out its production options at variant $j$: it evaluates (9) or (10), depending on the firm type. Eqs (9) and (10) also take account of the number of firms already active at the targeted variant.

Choice (iv) is considering abandoning a product variant. High niche spanning costs may prevent the firm from spreading its offering across several product variants. Thus, firms may increase profits by cutting costs by decreasing their niche width. The expected profit under

this choice is:

$$\tilde{\pi}''''_{i,t+1} = \sum\nolimits_{j\in H_{i,t}-\{d\}} \tilde{\pi}_{i,j,t+1} - N s_{H_i-\{d\},t}, \qquad (18)$$

where $d$ is the variant to abandon–i.e., the one with the least expected profit at the next simulation step.

## 3.5 Price behavior

In price equation $P = a - b_{j,t}Q$, we set intercept $a$ equal to total demand $M$. Slope $b_{j,t}$ is set so that market supply equals demand at zero price. The simulation begins with a single product variant that captures all $M$ demand. $M$ is redistributed equally among existing variants as the number of variants grows, and the price range is preserved after subsequent rescaling across variants. Firms cannot, in general, estimate the impact of the demand redistribution when a new variant is created. Thus, when considering expansion into a new variant, they take the recently observed slope as an input estimate for the next time step price equation [see Eqs (15) and (16)].

## 4 Simulation and results

We briefly summarize the core aspects of the simulation setting. Without loss of generality, we have a two-dimensional frame space with $11^2 = 121$ possible product variants, and with attribute values ranging from 0 to 10 along both axes. This set of values for the number of possible variants is, of course, arbitrary, but offers a wide range of possibilities to explore evolutionary dynamics. Firm types ($\alpha$-type and $\beta$-type) are assigned to the population $F$ of firms with equal probability, where each firm preserves its type during the simulation. Trials revealed that 200 time-steps per simulation run is usually enough to arrive at close-to-stable outcomes. In line with our focus on product variety impact on firm performance, we considered scenarios varying values along three variables: the number of firms ($F$), product variant opening cost ($K$), and product niche spanning cost ($N$). The value ranges of the scenarios have been determined through intensive experimentation. Experiments with $N$ and $K$ values outside these ranges have led to a trivial space evolution. Setting too high values does not allow the market to evolve. Setting too low values eliminates product variant creation and niche size differentials between firm types. Likewise, experiments with very few firms did not provide enough data to compare firm types, while having too many firms led to vanishing scale advantage differences. All the denotations of variables, parameters and their value ranges can be found in the supplementary material (S1 File).

We used a Latin hypercube sampling (LHS; see [53]) to reduce the number of parameter value combinations between the chosen variables without substantial information loss. LHS is an optimization procedure used in simulation to cover the whole parameter space with the minimum possible number of combination points. Through the LHS design, we considered 20 different scenarios (see their parameter listing in the supplementary material, S1 File). To guarantee orthogonality between parameters, we randomly generated 10,000 designs and selected the one with lowest average absolute pairwise correlation. To decide on the number of simulation runs per design point, we computed the coefficient of variation over the following variables per time-step: market concentration (Herfindahl) index, average Cournot competition profit per firm type, and fraction dimensionality. Coefficients $c_{\beta,j}$ and $e_j$ have been generated by assuming uniform distributions and according to their value constraints ($a - c_{\beta,j} > 0$, $c_{\beta,j} > e_j M$ and $e_j < b_j/2$; cf. Subsection 3.3).

The simulation starts with a single product variant offering located at the center of the frame space. Thus, product niche overlap between firms is maximal at the outset, strongly facilitating horizontal differentiation.

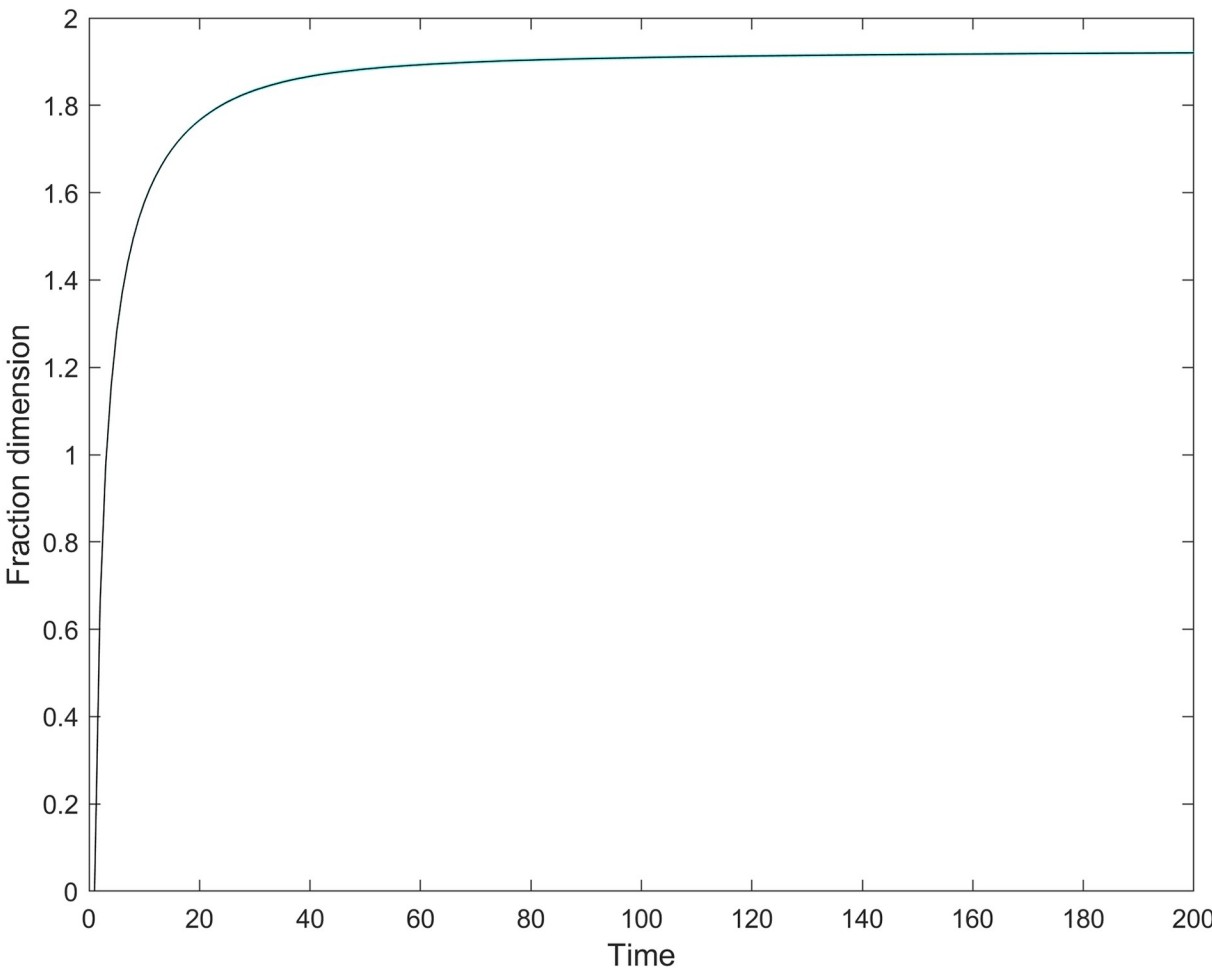

**Fig 8. The change of average fraction dimensionality.** The 95% confidence interval is negligible.

Fig 8 displays that following a steep initial increase, fraction dimensionality reaches a close-to-stable state at about $t = 40$, $DIM \approx 1.87$; from then on, $DIM$ converges very slowly to its maximum value 2, typically without reaching full saturation. Hence, the simulation window can be divided into two characteristically different stages. In the shorter initial phase, firms can reap first-mover advantages by introducing new product variants, extending their niches to empty cells. Their moves decrease the initially large niche overlap between firms, and so decrease competition intensity. Profit conditions temporally improve, in some extent compensating for the one-time product introduction costs as well as for the costs of maintaining a broader product niche. In the longer second phase, the market is more or less saturated with product variants; then, firms may increase their efficiency by stealing demand from their less efficient competitors.

Fig 9 illustrates the frequency of product introduction along the simulation time span. Results reveal that the speed of new variant creation reaches a peak of about 5.5 variants per step quite early, at around time-step 5, when $DIM \approx 1.37$ (see average $DIM$ values per simulation step in the supplementary material, S1 File). From that time-step on, variant creation drops drastically. Average new variant creation approximates zero way before reaching half simulation time. Dragging consumers to new product variants becomes less profitable, partly

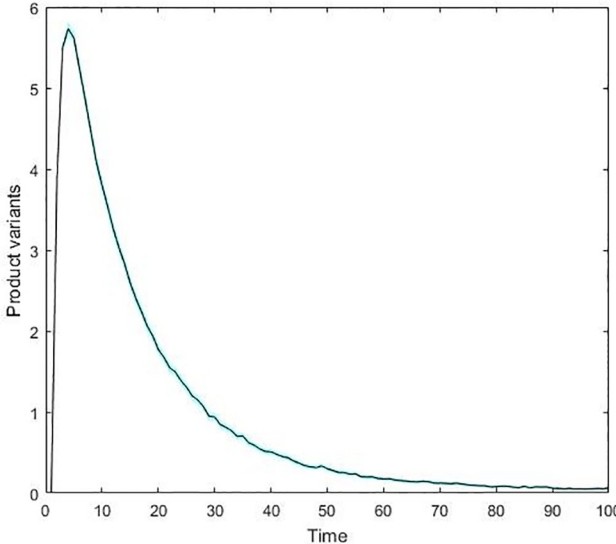

**Fig 9. The average number of created variants per simulation step.** The 95% confidence interval is negligible. Firms find it profitable to open new variants at the onset of the market; the aggregated actions trigger an explosion of variant creation per time-step that peaks at around 5.5 variants, on average. These aggregated actions of firms spread out the demand across the attribute space, implying that further incentives to open new variants sharply decline over time.

because the constant total demand thins out with the increasing number of product variants in place. Efficiency improvement through niche expansion becomes marginal; firms already securing an efficient producer status dominate at the detriment of the least efficient.

Market concentration shows an increasing trend in the longer run, but still remaining relatively low (Fig 10). In general, market concentration is more likely to increase under the presence of a highly heterogeneous set of firms. For instance, a market with a set of 50 equally-sized firms (total homogeneity) depicts a lower concentration figure $\left(50 * \left(\frac{1}{50}\right)^2 = 0.02\right)$ in comparison with a market with the same population size where 5 firms have 90% and the remaining 45 firms own 10% $\left(5 * \left(\frac{0.9}{5}\right)^2 + 45 * \left(\frac{0.1}{45}\right)^2 = 0.162\right)$. But now, firm heterogeneity is constrained by model design, allowing for the appearance of only two firm types with given growth characteristics. Initially, concentration sharply declines, in parallel with intense new product creation. Concentration begins its long increasing trend as the speed of variant creation starts to decline ($DIM \approx 1.5$, at a time-step below 10); later, it approximates a stable maximum just about the same time when the market approaches saturation with variants ($DIM \approx$ 1.9; compare Figs 8 and 10). This coincidence suggests that opening new, uncontested product variants is the way to soften competition, implying a more even market share distribution in the pre-saturation phase.

Next, we address the selective impact of increasing fraction dimensionality on $\alpha$-type and $\beta$-type firms. We find that the temporal pattern of the induced differences varies along variables; for example, while niche width differentials become even more pronounced with time (cf. Fig 11a), performance differences feature a non-monotonic, first increasing and then vanishing, pattern (cf. Fig 14). The scale advantage-holding $\alpha$-type firms tend to maintain somewhat larger niches than $\beta$-type firms in the mature market phase (Fig 11a and 11b); the difference is significant at a 95% confidence level in the market saturation phase. Fig 11a also reveals that the niche widths differentials are negligible during the initial market phase. Then, the

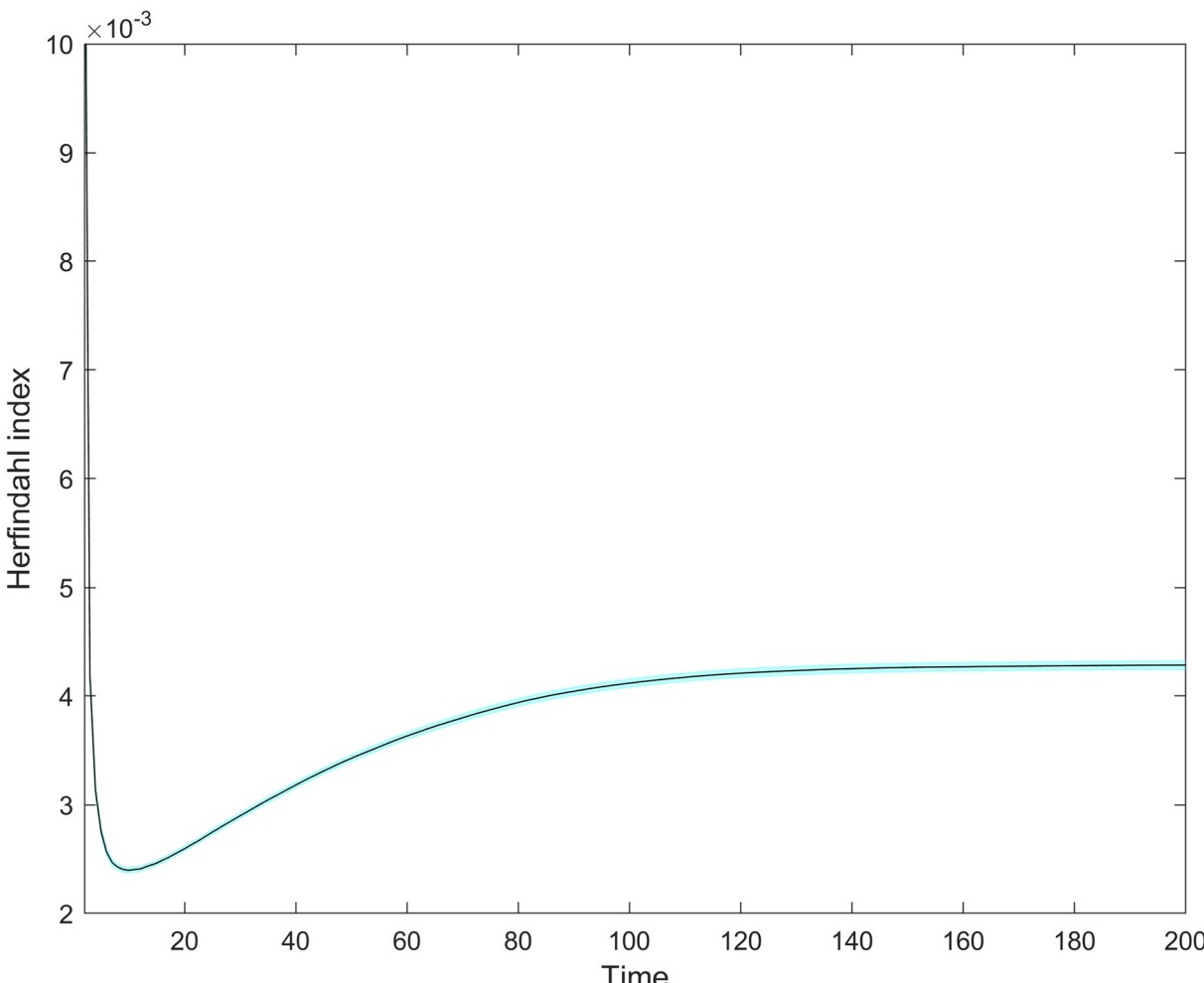

**Fig 10. The average behavior of the Herfindahl concentration index over time.** The 95% confidence interval is negligible. Market concentration first declines as a consequence of variant opening, which leads to firm distribution across the space and relief of direct competition. When the market gradually approaches saturation, variant opening declines and direct competition rises, favoring those with enough scale advantage to increase their market shares. Although qualitatively expected in terms of initial decline and further increase, market concentration values remain low due to model design decisions of only two firm types and a constant firm population.

availability of available product positions is high, and total demand distributes over a still low number of product variants. This abundant demand at easily accessible positions compensates for the one-time $K$ product opening costs. As demand density is thinning out with the increasing number of variants, maintaining a broad niche becomes more difficult. The scale economy advantages of $\alpha$-type firms can somewhat compensate for these losses. Fig 11a also indicates that the aggregate niche size difference between $\alpha$-type and $\beta$-type firms is getting larger after steps 80–90. The fraction dimensionality values corresponding to this range (S1 File) indicate, nonetheless, near saturation. Fig 12 confirms that the majority of these openings are made by $\alpha$-type firms. Scale advantages play a more pronounced role in product introduction as the market gets saturated with variants.

Fig 12 also indicates that product introduction becomes extremely rare in the second market phase. But this is at odds with the fact that there are still several not yet opened positions

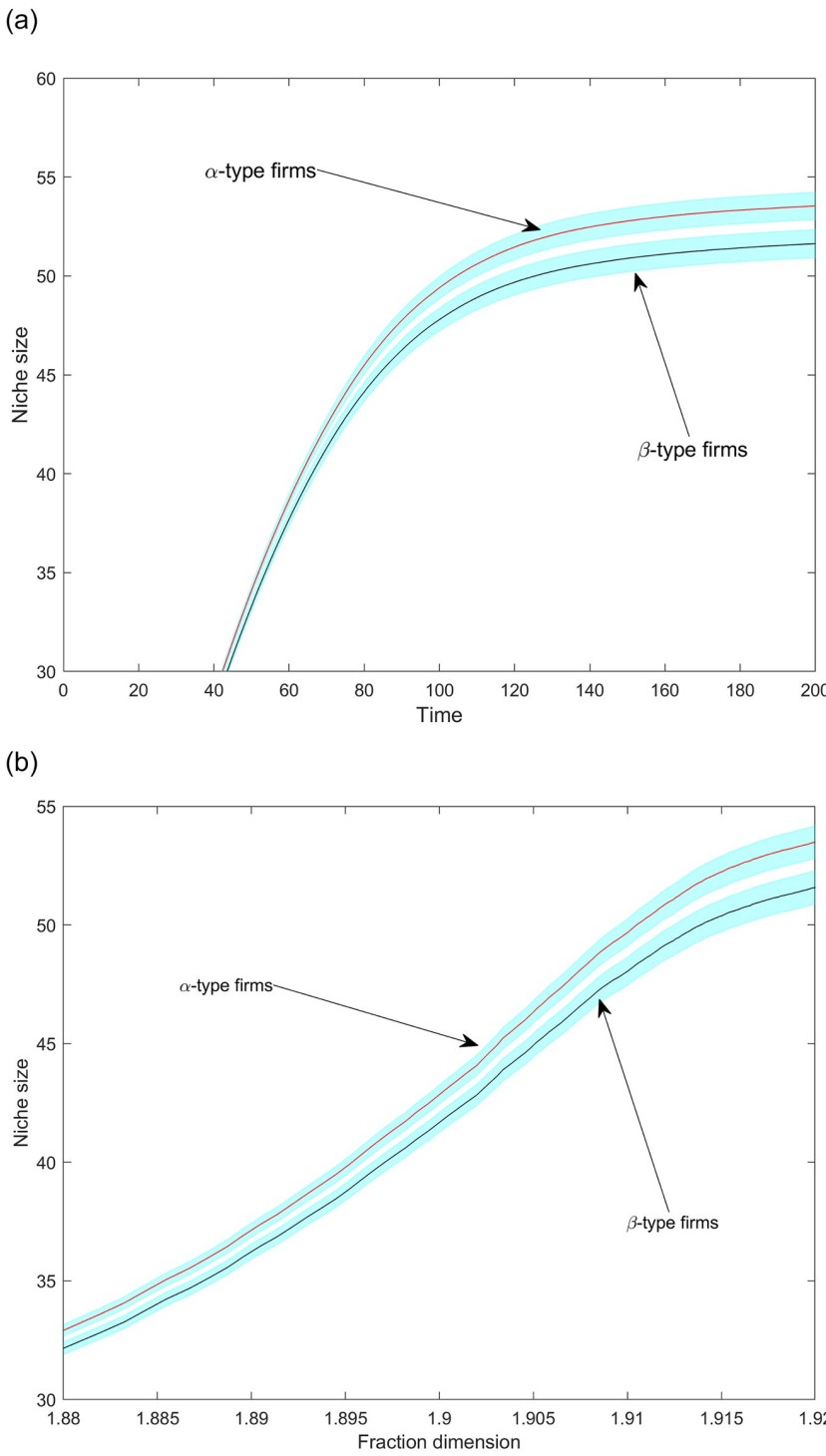

**Fig 11. The average aggregate niche size per firm type over time (a) and as a function of fraction dimension (b).**
Shaded regions are 95% confidence intervals.

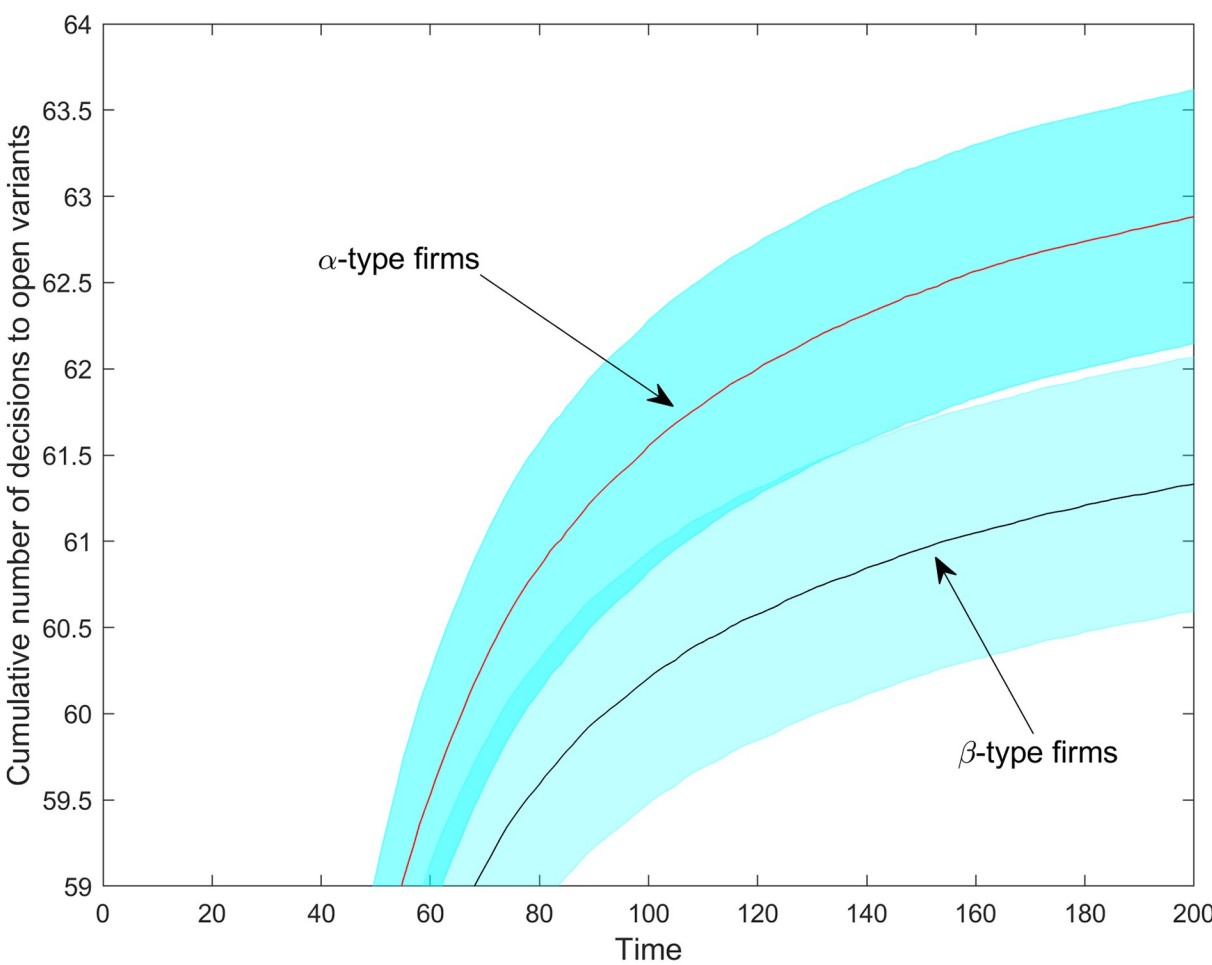

**Fig 12. The average cumulative number of attempts to open new variants per firm type as a function of fraction dimensionality.** Shaded regions are 90% confidence intervals.

around; for example, almost 20% of the cells are yet available at half simulation time 100 ($DIM$ = 1.9098). So, there must be at least one more constraining effect on product niche expansion we still have to identify.

Product landscapes are usually patchy. The still available product cells can be isolated and so firms aiming to open a cell have to trespass a barrier of in-between occupied cells first. [9] argue that beyond its size, the geometry of the product niche also affects its sustainability. A niche is convex if it contains all positions along the line between its two arbitrary positions. A convex niche has no *caveat*s, and other things being equal, it is better for the firm than a concave niche with the same number of occupied cells. This is because the neighboring positions in the niche used to stand for similar products, a fact that can give rise to scope economies at joint production. Speaking about the similarity of neighboring cells presumes that the product dimensions that span the frame space have ordering scales at least. Note furthermore that our 'low resolution' frame space has only 11 x 11 cells, so the convexity definition based on connecting lines can only be applied as an approximation.

The firms in our model setting thrive for niche convexity when they scout for potential new product positions neighboring to their extant offerings (see option ii in Subsection 3.4).

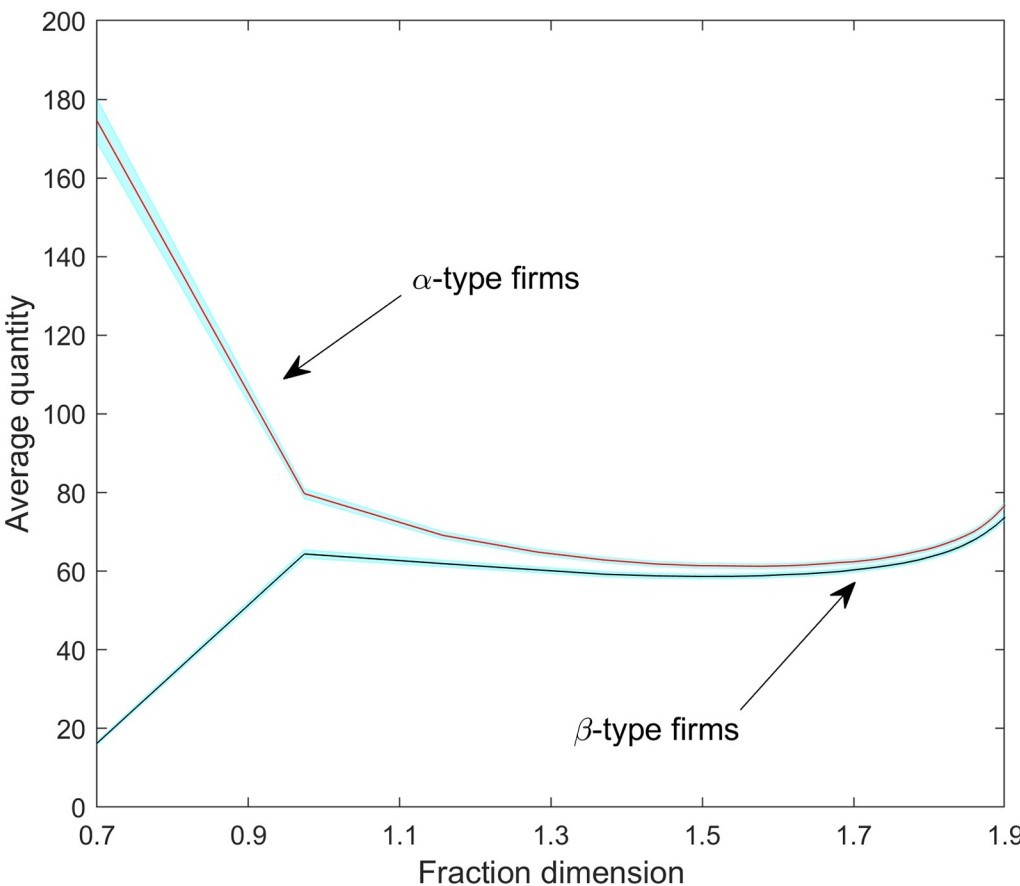

**Fig 13. The average multi-variant Cournot quantity per firm type as a function of fraction dimensionality.** Shaded regions are 95% confidence intervals.

In the long run, niche differentials between firm types stabilize (Figs 11 and 12). The rare product introduction makes the rivalry for product positions a zero-sum game. However, these do not exclude positioning changes (see Fig 11b where significant niche size changes occur–between 30 and 55 variants–within a narrow range values of fraction dimension–1.88 to 1.92). How do firms occupy product positions once belonged to another firm: peacefully or by competitive exclusion? Do firms force out others from certain domains or just pick up products abandoned voluntarily before? The next two figures will help answering these questions.

Fig 13 corroborates that scale advantages help $\alpha$-*type* firms to reap first-mover advantages at low dimensionality stages. Although $\alpha$-*type* firms manage to maintain somewhat higher quantities than $\beta$-*type* firms along the whole dimensionality spectrum, the difference is significantly reduced as dimensionality increases. Fig 13 also indicates that increasing dimensionality mainly helps to increase demand for only one firm type: the $\beta$-*type*. However, $\alpha$-*type* firms accumulate higher profits (see below). $\alpha$-*type* firms maintain their larger niches and cumulative utility values through price effects (and not due to a demand gain) in the market saturation phase. By inspecting again Fig 12, we additionally conclude that actions of $\alpha$-*type* firms in terms of variant opening might help to increase demand for $\beta$-*type* firms. This finding

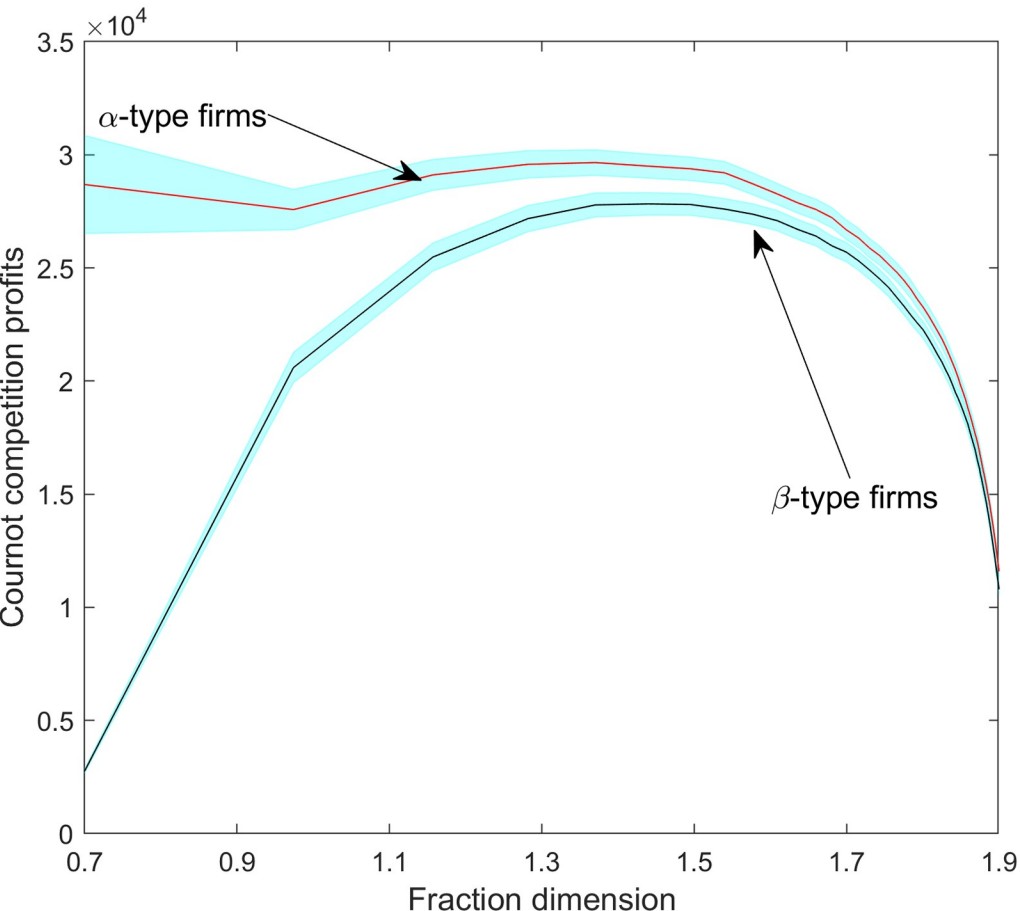

**Fig 14. The average Cournot competition profits (i.e., without including variant opening costs) per firm type as a function of fraction dimensionality.** Shaded regions are 95% confidence intervals.

facilitates a better understanding of the impact of product proliferation, as well as firm-level characteristics and firm interaction [54].

Fig 14 displays the generated profits of Cournot competition rounds from *DIM* = 0.7, which corresponds to the phase when firms decide to open or suppress variants. Fig 14 reveals that *α-type* firms have an early advantage in terms of profits as well. But again, *β-type* firms, without scale economies, reduce their disadvantage as the space becomes saturated with product variants. From about *DIM* = 1.4 (5–7 time-steps), profits turn heavily decreasing for both firm types. From about *DIM* = 1.8 (circa 25 time-steps), it is not possible to statistically differentiate between the profit performance of the two types. These findings are in line with a core conclusion of the *resource-based theory of markets* [48]: the increasing number of product variants can endogenously generate scope diseconomies, undermining scale effects and thus *improving* the relative position of firms without scale advantages. In our case, this relative improvement takes place with steeply decreasing, though still positive, profits. Note, however, that even though fraction dimensionality erodes Cournot profits for *α-type* firms, they are still ahead of *β-type* firms in terms of cumulative profits (Fig 15).

Having similar and mutually decreasing Cournot profits indicates that *Red Queen* competition might take place in the market [55, 56]. *Red Queen* competition theory is adopted from evolutionary biology [57], positing that competing firms must mutually improve their

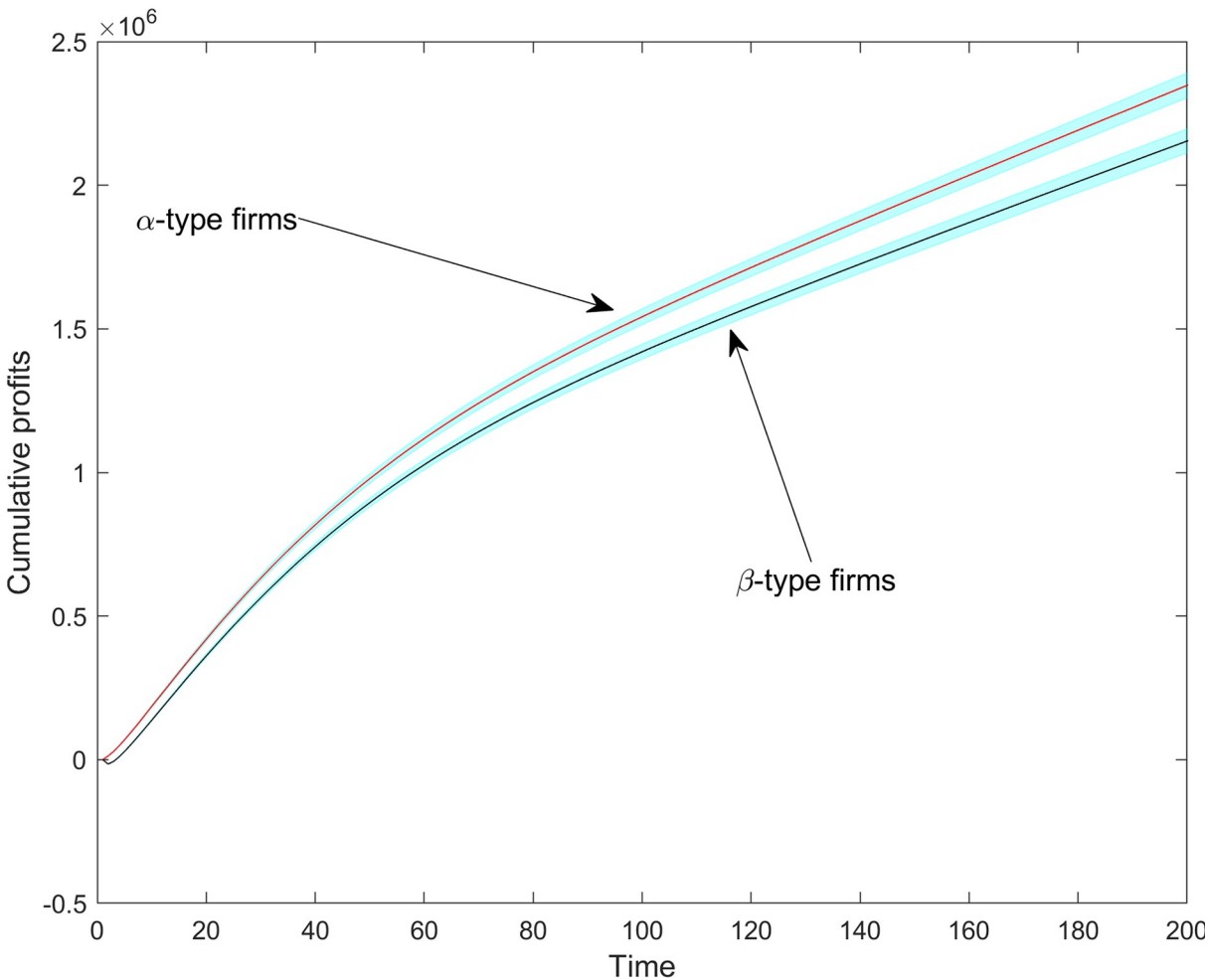

**Fig 15. The average cumulative profits per firm type over time.** Shaded regions are 95% confidence intervals.

efficiency in the course of ongoing competition just to maintain their market positions relative to their competitors. In our model, efficiency can improve by better choosing the product portfolio. As getting even both in profits and in production quantities (Figs 13–14), our two firm types represent similar market strengths. This might suggest that firms adduce product variants in the mature market phase not so much by robbing positions from others, but by possibly re-using positions abandoned by others.

## 5 Conclusions

The added value of computational approaches is getting increasingly acknowledged in the study of market interactions [58]. Our model presents a specific instance of co-evolution between the product space and the populating firms in which increasing fraction dimensionality affects firms with and without scale economies differently, in a nonlinear fashion. The competitive outcomes we discussed emerge in a framework with firms being agnostic about other firms' actions, as well as about the type of those other firms with which they interact. Competition is not only face-to-face, but also diffuse: actors experience competition intensity changes

through the changing availability of demand at the product positions. Firms are myopic in a spatial and temporal sense as well: they consider conditions in the one-cell neighborhood of their realized niches and make decisions always only one time step ahead.

The simulations reveal a two-stage market dynamics. In the shorter initial phase, new market positions are yet plenty so that *α-type* firms with scale economies can reap easy, though transient, first-mover advantages. The market is approaching, although not reaching, full saturation with product variants in the second market phase. Changes are gradual showing robust tendencies in this much longer phase. As fraction dimensionality approximates its maximum, the scenery becomes increasingly similar to a product space with two integer dimensionalities. Accordingly, the findings of integer dimensionality model settings are expected to occur. Our simulation reproduces the well-documented trade-off between organizational niche breadth and fitness [9, 59] with respect of this second market phase. Since total market demand is fixed, the demand in the active cells thins out as a result of the slowly ongoing product variant introduction. As *α-type* firms keep on seeking to maintain exclusive demand pockets through the creation of new product variants, they gradually weaken their scale advantages: their limited innovative capacities spent on new product variants is getting smeared thinly along their broadening product niches. This improves the relative position of the rest; so *β-type* firms experience a somewhat lessening competitive pressure with increasing fraction dimensionality.

The event history of the second market phase is also in line with the predictions and empirical findings of the ecological theories of markets on narrow-niche firms' improving performance relative to large scale firms as concentration rises [8, 59, 60]. Explanations based on increasing integer attribute space dimensionality have also been proposed for the oftentimes observed performance improvement of small specialist firms in maturing markets [19, 29]. Considering the need to understand how firms create economic opportunities by shaping the space they compete in, our model adheres to the recent literature on competitive landscape dynamics [61, 62]. A novelty of our work is explaining similar structural outcomes with reference to another phenomenon: the market's increasing saturation with product varieties. Smaller specialist organizations, impersonated by *β-type* firms, experience a competitive relief relative to the larger, scale-driven *α-type* firms as fraction dimensionality increases. We find, moreover, that this increasing fraction dimensionality can also take place in parallel with increasing market concentration. Our model also reveals a new form of myopic firm behavior that manifests in favoring attractive short-run considerations with potentially negative strategic consequences. When *α-type* firms introduce a new product variant, they modify the local tissue of the attribute space by activating a new cell. Doing so, they are paving the way for their *β-type* competitors to make offerings for the demand at these cells in the longer run.

Our simulations explore how competitive market interactions may evolve with firms only reacting to the demand availability in their vicinity. The mathematical concept of fraction dimensionality has become a key indicator in this endeavor with its change indicating shifting competitive conditions. A next step in this line of work is adding additional features of face-to-face competition, and so additional reactive strategic considerations, to the picture. This includes letting firms perceive the actual properties (market share, type, niche breadth, et cetera), the spatial positioning, and also the recent actions of their competitors, and letting them (re)act accordingly. Global market indicators like total demand and supply, or the saturation level with product varieties, might serve as crucial information for individual strategy formation. Fraction dimensionality can serve as an explanatory variable at the exploration of even more complex firm interactions in co-evolutionary contexts.

## Supporting information

**S1 File.**
(DOCX)

## Author Contributions

**Conceptualization:** César García-Díaz.

**Formal analysis:** César García-Díaz, Gábor Péli, Arjen van Witteloostuijn.

**Investigation:** César García-Díaz, Gábor Péli, Arjen van Witteloostuijn.

**Methodology:** César García-Díaz, Gábor Péli, Arjen van Witteloostuijn.

**Resources:** Arjen van Witteloostuijn.

**Writing – original draft:** César García-Díaz, Gábor Péli, Arjen van Witteloostuijn.

**Writing – review & editing:** César García-Díaz, Gábor Péli, Arjen van Witteloostuijn.

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
