## [Decision Letter · Decision Letter 0]

5 Mar 2020

PONE-D-19-34089

The Coevolution of the Firm and the Product Attribute Space

PLOS ONE

Dear Dr. García-Díaz,

Thank you for submitting your manuscript to PLOS ONE. After careful consideration, we feel that it has merit but does not fully meet PLOS ONE’s publication criteria as it currently stands. Therefore, we invite you to submit a revised version of the manuscript that addresses the points raised during the review process.

We recommend that it should be revised taking into account the changes requested by the reviewers. Since the requested changes includes Major Revision, the revised manuscript will undergo the next round of review by the same reviewers.

We would appreciate receiving your revised manuscript by Apr 19 2020 11:59PM. To enhance the reproducibility of your results, we recommend that if applicable you deposit your laboratory protocols in protocols.io, where a protocol can be assigned its own identifier (DOI) such that it can be cited independently in the future. For instructions see: http://journals.plos.org/plosone/s/submission-guidelines#loc-laboratory-protocols

We look forward to receiving your revised manuscript.

Kind regards,

Baogui Xin, Ph.D.

Academic Editor

PLOS ONE

Reviewers' comments:

Reviewer's Responses to Questions

**Comments to the Author**

1. Is the manuscript technically sound, and do the data support the conclusions?

Reviewer #1: Yes

Reviewer #2: Yes

2. Has the statistical analysis been performed appropriately and rigorously? 

Reviewer #1: N/A

Reviewer #2: Yes

3. Have the authors made all data underlying the findings in their manuscript fully available?

Reviewer #1: Yes

Reviewer #2: Yes

4. Is the manuscript presented in an intelligible fashion and written in standard English?

Reviewer #1: Yes

Reviewer #2: Yes

5. Review Comments to the Author

Reviewer #1: The paper develops an alternative way to analyze firms competition which differs from the traditional setup in which the dimensionality of the product attribute space is given. The focus of the analysis is to make the product attribute space endogenous depending on the firms' actions which, in turn, influence the firms' behavior. The analysis is conducted within a networked Cournot competition framework and a key role is played by the fraction dimensionality which reflects the saturation of the space with product varieties. The findings of the analysis reveal that, even a larger number of new products is introduced by firms with scale economies, the performance gap with firms that do not benefit from scale economies reduces as the fraction dimensionality increases.

Overall the paper is pretty clear and well written and deals with an interesting topic related to firms competition, which is a relevant subject in the industrial sector analysis, and it may deserve publication in Plos One.

The authors may take into account the comments below:

1. In equation 3, I think a subscript i is missing after pi. Then authors may want to explain why they select a particular value of 0.2 for the mobility rate and those values for the demand parameters: is it only for an explanatory purpose? The conclusion at the end of sec. 3.1 depends on the parameter choice: is it possible to generalize it and find a condition under which profits are positive and the profit of scenario B is higher than A? What are product coordinate values in fig.7 and what do they represent?

2. Figure 9 is not easy to read: maybe the reader would gain more insight from it if the fraction dimension was put on the vertical axis and the description of the figure at pag. 19 was enlarged by adding more economic intuition.

3. The description of fig. 10 should also be improved when explaining the concentration issue.

Reviewer #2: Under the framework of product attribute space, the competition among firms is studied in this paper. The authors first introduce the concept of fractional dimension to describe attribute space. And then they build their own models based on the traditional Cournot model and the networked Cournot model, respectively. At last, the numerical simulations are carried out by using the built computational models, to enclose the firms’ performances when new product variants are introduced. The model in this article is original and the research results are quite interesting. However, I still have several comments about this research, which can be found in the appendix.

6. PLOS authors have the option to publish the peer review history of their article (what does this mean?). If published, this will include your full peer review and any attached files.

Reviewer #1: No

Reviewer #2: Yes: Wei Zhou

---

## [Author Response · Author response to Decision Letter 0]

25 Apr 2020

Following the directions of the editor, we have:

(i) Uploaded a rebuttal letter responding to each point raised by the reviewers. This letter is labeled 'Response to Reviewers' [Response_to_referees_April_24_2020.docx] in the attached files.

(ii) Uploaded a marked-up copy of the manuscript that highlights changes made to the original version. This file is labeled 'Revised Manuscript with Track Changes' [Revision_File_track_changes_April_24_2020.docx] in the attached files.

(iii) Uploaded an unmarked version of the revised paper without tracked changes. This file should is labeled 'Manuscript' [Revised_manuscript_April_24_2020.docx] in the attached files.

We also want to acknowledge in the paper that this project has been financially supported by the “Vicerrectoria de Investigación / Facultad de Ciencias Económicas y Administrativas” of Pontificia Universidad Javeriana (Project ID: 9546).

---

## [Editor Report · Decision Letter 1]

18 May 2020

The Coevolution of the Firm and the Product Attribute Space

PONE-D-19-34089R1

Dear Dr. García-Díaz,

We are pleased to inform you that your manuscript has been judged scientifically suitable for publication and will be formally accepted for publication once it complies with all outstanding technical requirements.

With kind regards,

Baogui Xin, Ph.D.

Academic Editor

PLOS ONE
---

## [Editor Report · Acceptance letter]

28 May 2020

PONE-D-19-34089R1 

The Coevolution of the Firm and the Product Attribute Space 

Dear Dr. García-Díaz:

I am pleased to inform you that your manuscript has been deemed suitable for publication in PLOS ONE. Congratulations! Your manuscript is now with our production department. 

With kind regards,

on behalf of

Professor Baogui Xin 

Academic Editor

PLOS ONE